# Adaptive evolution among cytoplasmic piRNA proteins leads to decreased genomic auto-immunity

Luyang Wang[1], Daniel A. Barbash[2], Erin S. Kelleher[1]*

1 Dept. Biology & Biochemistry, University of Houston, Houston, Texas, United States of America, 2 Dept. Molecular Biology & Genetics, Cornell University, Ithaca, New York, United States of America

* eskelleher@uh.edu

**Data Availability Statement:** Data have been deposited in the NCBI sequence read archive under PRJNA494103 (https://www.ncbi.nlm.nih.gov/bioproject/PRJNA494103).

## Abstract

In metazoan germlines, the piRNA pathway acts as a genomic immune system, employing small RNA-mediated silencing to defend host DNA from the harmful effects of transposable elements (TEs). Expression of genomic TEs is proposed to initiate self regulation by increasing the production of repressive piRNAs, thereby "adapting" piRNA-mediated control to the most active TE families. Surprisingly, however, piRNA pathway proteins, which execute piRNA biogenesis and enforce silencing of targeted sequences, evolve rapidly and adaptively in animals. If TE silencing is ensured through piRNA biogenesis, what necessitates changes in piRNA pathway proteins? Here we used interspecific complementation to test for functional differences between *Drosophila melanogaster* and *D. simulans* alleles of three adaptively evolving piRNA pathway proteins: Armitage, Aubergine and Spindle-E. In contrast to piRNA-mediated transcriptional regulators examined in previous studies, these three proteins have cytoplasmic functions in piRNA maturation and post-transcriptional silencing. Across all three proteins we observed interspecific divergence in the regulation of only a handful of TE families, which were more robustly silenced by the heterospecific piRNA pathway protein. This unexpected result suggests that unlike transcriptional regulators, positive selection has not acted on cytoplasmic piRNA effector proteins to enhance their function in TE repression. Rather, TEs may evolve to "escape" silencing by host proteins. We further discovered that *D. simulans* alleles of *aub* and *armi* exhibit enhanced off-target effects on host transcripts in a *D. melanogaster* background, as well as modest reductions in the efficiency of piRNA biogenesis, suggesting that promiscuous binding of *D. simulans* Aub and Armi proteins to host transcripts reduces their participation in piRNA production. Avoidance of genomic auto-immunity may therefore be a critical target of selection. Our observations suggest that piRNA effector proteins are subject to an evolutionary trade-off between defending the host genome from the harmful effect of TEs while also minimizing collateral damage to host genes.

**Funding:** This research was supported by the Cornell Center for Comparative and Population Genomics and the University of Houston Division of Research. E.S.K. was supported by the Cornell Center for Comparative and Population Genomics (http://3cpg.cornell.edu/), an NIH National Research Service Award (F32GM090567-01) (https://www.nih.gov/), and NSF-DEB 1457800 (https://www.nsf.gov/) (to E.S.K.). Luyang Wang was supported by NSF-DEB 1457800 (to E.S.K.). D.A.B. was supported by NIGMS R01GM074737. The funders had no role in study design, data collection and analysis, decision to publish, or preparation of the manuscript.

**Competing interests:** The authors have declared that no competing interests exist.

## Author summary

Transposable elements are mobile fragments of selfish DNA that burden host genomes with deleterious mutations and incite genome instability. Host cells employ a specialized small-RNA mediated silencing pathway, the piRNA pathway, to act as a genomic immune system suppressing the mobilization of TEs. Changes in genomic TE content are met with rapid changes in the piRNA pool, thereby maintaining host control over transposition. However, piRNA pathway proteins—which enact piRNA biogenesis and silence target TEs—also evolve adaptively. To isolate forces that underlie this adaptive evolution, we examined functional divergence between two *Drosophila* species for three adaptively evolving piRNA pathway proteins. To our surprise, we found very few differences in TE regulation, suggesting that evolution has not generally acted to enhance control of TE parasites. Rather, we discovered interspecific differences in the regulation of host mRNAs for two proteins, which suggested that proteins evolve to avoid off-target silencing of host transcripts. We propose that the avoidance of such "genomic autoimmunity" is an important and underappreciated force driving the adaptive evolution of piRNA proteins.

## Introduction

Transposable elements (TEs) are ubiquitous mobile genetic entities, whose unrestricted propagation can cause deleterious insertional mutations and chromosome rearrangements, and are often associated with cancer and sterility [1–4]. TE regulation is therefore essential, especially in germline cells, where TE insertions and associated mutations can be transmitted to the next generation. In metazoan germlines, regulation of TE transcripts is enacted by a small RNA silencing pathway, the PIWI-interacting RNA pathway (piRNA pathway), in which piRNAs complexed with PIWI-clade Argonaute proteins target complementary TEs for post-transcriptional and transcriptional silencing [5].

Host genomes are often parasitized by multiple TE families, which change rapidly in their presence and abundance [6–9]. The control of TE transcripts by complementary piRNAs may facilitate adaptation to genomic TEs through changes in piRNA species [10,11]. Surprisingly, however, the protein components of the piRNA pathway that enact piRNA biogenesis and enforce TE silencing also evolve adaptively in diverse metazoan lineages [12–16]. Evidence for adaptive evolution of piRNA pathway proteins is particularly strong in *Drosophila* [12–15,17], which has also emerged as a work horse for uncovering the mechanisms of piRNA-mediated silencing [reviewed in 18]. For example, a recent meta analysis including both *D. melanogaster* and *D. pseudoobscura* revealed that 22 of 26 piRNA pathway proteins exhibit significant signatures of adaptive protein evolution in one or both species [14].

Adaptive evolution of piRNA effector proteins is proposed to arise from an evolutionary arms race between TEs and host silencing machinery [reviewed in 19]. In the simplest scenario, effector proteins evolve adaptively in order to restore silencing of newly invading or escaper TE families. Alternatively, if TEs "fight back" by encoding RNA or protein antagonists of host silencing machinery, piRNA pathway proteins could evolve adaptively to escape TE antagonism [20]. Finally, piRNA proteins may evolve adaptively to avoid "genomic auto-immunity" in the form of off-target silencing of host genes [19,21]. Uncovering which of these selective forces drives the adaptive evolution of piRNA effector proteins requires elucidating the resulting functional consequences of piRNA-effector-protein divergence. For example, adaptive evolution among transcriptional silencers has led to incompatibilities between alleles of interacting proteins from different species, with dramatic consequences for piRNA

production and TE control [20,22,23]. In particular, functional changes in Rhino are proposed to reflect evolutionary escape from a TE-encoded antagonist [20].

Here, we broaden our understanding of the functional consequences of adaptive evolution among *Drosophila* piRNA effector proteins by examining three additional essential piRNA pathway components that play critical roles in piRNA maturation and post-transcriptional silencing [24–30]: Armitage (Armi), Aubergine (Aub) and Spindle-E (Spn-E). This work significantly extends a preliminary analysis of Aub divergence [31]. Aub is a Piwi-clade Argonaute protein which, guided by piRNAs, enacts post-transcriptional silencing of sense TE-derived mRNAs [24]. Aub cleavage also feeds forward the ping-pong amplification cycle, a core mechanism for the maturation of both sense and antisense piRNAs that also requires Spn-E [25,26,28,32]. Distinct from both Aub and Spn-E, Armi binds to antisense piRNA precursors to facilitate their sequential cleavage by the nuclease Zucchini in an alternate biogenesis mechanism referred to as "phasing" [29,33–37]. The loci encoding Aub, Spn-E and Armi all exhibit adaptive evolution along the lineage leading to *D. melanogaster*, *D. simulans* or both, yet the underlying evolutionary force(s) remain unknown [13,15].

To isolate diverged functions of these adaptively evolving proteins, we performed interspecific complementation, in which we compared the ability of *D. melanogaster* and *D. simulans* wild-type alleles to complement a *D. melanogaster* mutant background. While nuclear transcriptional silencers were previously demonstrated to exhibit dramatic interspecific divergence in TE regulation and piRNA production [20,23], we observed only minor allelic differences in both of these functions. Rather, we uncovered idiosyncratic differences in the regulation of a small handful of TEs, suggesting potential element-specific adaptations. We also observed that *D. simulans* alleles of *aub* and *armi* exhibit reduced efficiency of piRNA maturation in association with increased off-target regulation of host mRNAs. We propose that in contrast to nuclear transcriptional silencers, selection acts on cytoplasmic piRNA proteins to maximize their specificity to piRNA production and TE transcripts, while minimizing non-functional or deleterious interactions with host mRNA.

## Results

### Identifying functional divergence through interspecific complementation

Previous divergence-based analyses of Aub and Spn-E suggest that adaptive evolution is not confined to a particular functional domain but is dispersed throughout the proteins [12,15]. Consistent with these findings, we identified abundant amino-acid differences between *D. melanogaster* and *D. simulans* throughout Aub and Spn-E (Fig 1A). Armi does not exhibit strong evidence of positive selection in divergence-based tests, however, an excess of amino acid substitutions exists between *D. melanogaster* and *D. simulans*, which have likely arisen by positive selection in one or both lineages [15]. Similar to Aub and Spn-E, we observe that these amino acid differences are scattered throughout the protein, both inside and outside of functional domains (Fig 1A).

To isolate phenotypic differences between *D. melanogaster* and *D. simulans* alleles that result from adaptive evolution, we employed interspecific complementation, in which we compared the ability of *D. melanogaster* and *D. simulans* wild-type alleles to complement a *D. melanogaster* mutant background. For each selected piRNA protein, we generated and compared three genotypes: 1) a trans-heterozygous loss-of-function mutant, 2) the same mutant with a *D. melanogaster* genomic transgene rescue, and 3) the same mutant with a *D. simulans* genomic transgene rescue (S1 Fig). The transgenes include the complete genomic region from either *D. melanogaster* or *D. simulans*, including upstream and downstream sequences containing potential cis-regulatory elements. Transgenes were inserted into matched *attP* sites by

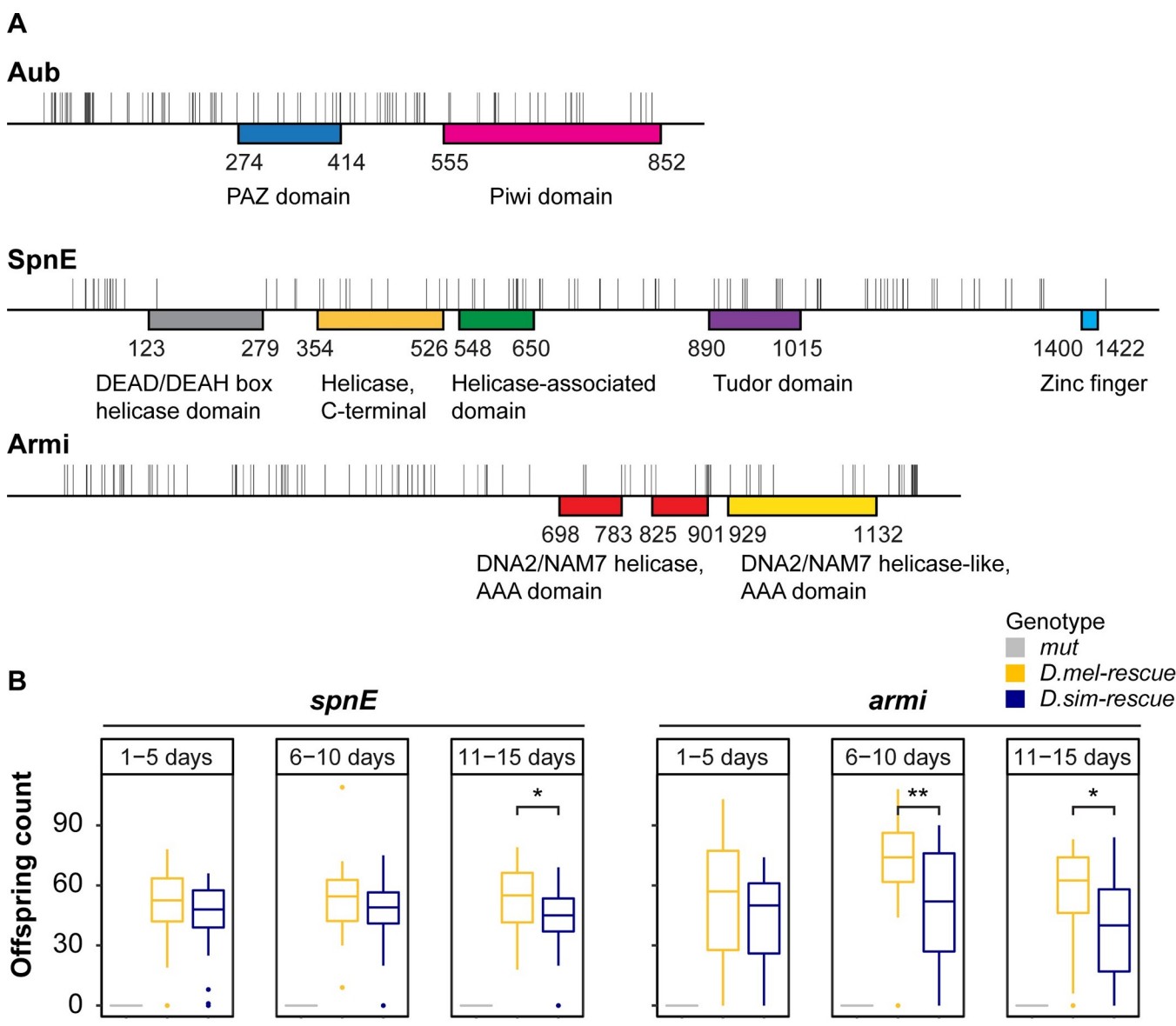

**Fig 1. Functional and sequence divergence in piRNA pathway proteins.** (A) Amino acid substitutions between *D. melanogaster* [38] and *D. simulans* [39] reference alleles are indicated as grey tick marks. Starting and ending amino acids for InterPro [40] annotated functional domains are indicated. (B) Female fertility for *D. melanogaster* and *D. simulans* transgenic rescues are compared for three different age classes. Females with the *D. simulans* spn-E transgene are significantly less fertile across the experiment (Repeated measures ANOVA, $F_{1,172} = 4.043$, $p < 0.05$) and at the third time point we measured (11–15 days, $t_{56} = 2.304$, $p < 0.05$). Females with the *D. simulans* armi transgene are significantly less fertile across the experiment (Repeated measures ANOVA, $F_{1,175} = 8.824$, $p < 0.01$) and at the second(06–10 days, $t_{57} = 3.0718$, $p < 0.01$) and the third time points (11–15 days, $t_{57} = 2.5915$, $p < 0.05$). Sample sizes are 25–35 females. * denotes $p \leq 0.05$. ** denotes $p \leq 0.01$.

ΦC31 integrase [41], in order to avoid variable position effects. Phenotypes for which the *D. simulans* alleles fail to fully complement the mutant, or otherwise differ between the alleles of the two species, point to diverged functions that are potential targets of adaptive evolution.

We first considered the effect of transgenic rescue on female fertility. Homozygosity or trans-heterozygosity for loss of function alleles in all three genes causes complete female sterility (Fig 1B, S1 Table), while heterozygotes are fertile [42]. For all three proteins, fertility is restored by transgenic rescues from the two species to different degrees, with *D. melanogaster*

transgenes conferring higher fertility than their *D. simulans* counterparts (Fig 1B, S1 Table) [31]. *Drosophila simulans* transgenes do not exhibit significantly reduced expression when compared to *D. melanogaster*, in fact for *spn-E* the *D. simulans* transgene exhibits a modest 22% increase in expression (S2 Fig). Therefore, reduced fertility effects in the presence of *D. simulans* transgenes most likely reflect amino acid sequence divergence in the encoded proteins.

### Idiosyncratic differences in TE regulation

To uncover molecular phenotypes that relate to fertility differences, we first examined whether *D. melanogaster* and *D. simulans* alleles differed with respect to TE transcriptional control and associated piRNA production using RNA-seq and small RNA-seq (S3 Fig, S2 Table). Notably, our *D. melanogaster* transgenic rescues down-regulated TE transcripts and up-regulated piRNA production similarly to heterozygotes, which are generally considered wild-type with respect to piRNA production and TE silencing (Fig 2, S4 Fig, S3 and S4 Tables). Transgenically expressed *D. melanogaster* proteins are therefore fully functional with respect to TE silencing and piRNA biogenesis.

Enhanced piRNA-mediated negative regulation of TEs is an obvious target of positive selection acting on piRNA pathway proteins. However, sense transcripts for the majority of TEs are not differentially expressed between the transgenic rescues for *armi* and *spn-E*, implying that negative regulation of TEs is largely conserved between species (Fig 2A). Similarly, the majority of TEs are not differentially expressed between *aub* transgenic rescues (non-stranded RNA-seq, Fig 2A). Nevertheless, despite an overall conservation of TE repression, we discovered idiosyncratic differences in regulation, in which individual TE families are more robustly silenced by the *D. melanogaster* or *D. simulans* allele. Unexpectedly, 5 out of 5 TE families whose transcript abundance differs between transgenic rescues for one of the three proteins are more robustly silenced by the *D. simulans* allele. While differences in TE copy number could arise between transgenes through backcrossing, thereby creating the appearance of differential regulation, this would be equally likely to result in increased or decreased expression in the *D. simulans* rescue for any given TE. Thus, the bias towards enhanced negative regulation by *D. simulans* is not consistent with the random segregation of polymorphic TE insertion alleles during backcrossing, but rather suggests a true increase in negative regulation by the *D. simulans* allele. In particular, the *tirant* LTR retrotransposon is more robustly silenced by the *D. simulans* allele of both *aub* and *spn-E*. Furthermore, we did not observe any systematic differences in expression for germline or soma-specific protein-coding genes between the transgenic rescues, strongly suggesting that the germline-to-soma ratio is equivalent between transgenic genotypes (S2 Fig). Differences in silencing of individual TE families therefore suggest lineage-specific coevolution with the host-regulatory machinery.

Divergence in TE regulation between *D. melanogaster* and *D. simulans* alleles could arise from differential production of complementary antisense piRNAs. Of 5 TE families that are differentially regulated between transgenic rescues for one of the proteins (Fig 2A), only *tirant* differential expression between *spn-E* transgenes is associated with a corresponding change in antisense piRNA abundance (S5 Fig). Furthermore, *tirant* antisense piRNAs were increased in the *D. melanogaster* transgene, which is not consistent with piRNA loss as the cause of increased TE expression. Similar to their antisense counterparts, the abundances of sense piRNAs, which are produced during post-transcriptional silencing of TE-derived mRNAs, are not systematically different between transgenic rescues for differentially regulated TEs (S5 Fig). Differences in TE negative regulation between alleles therefore occur independently of piRNA production. Furthermore, *D. simulans* and *D. melanogaster* alleles have very similar effects on

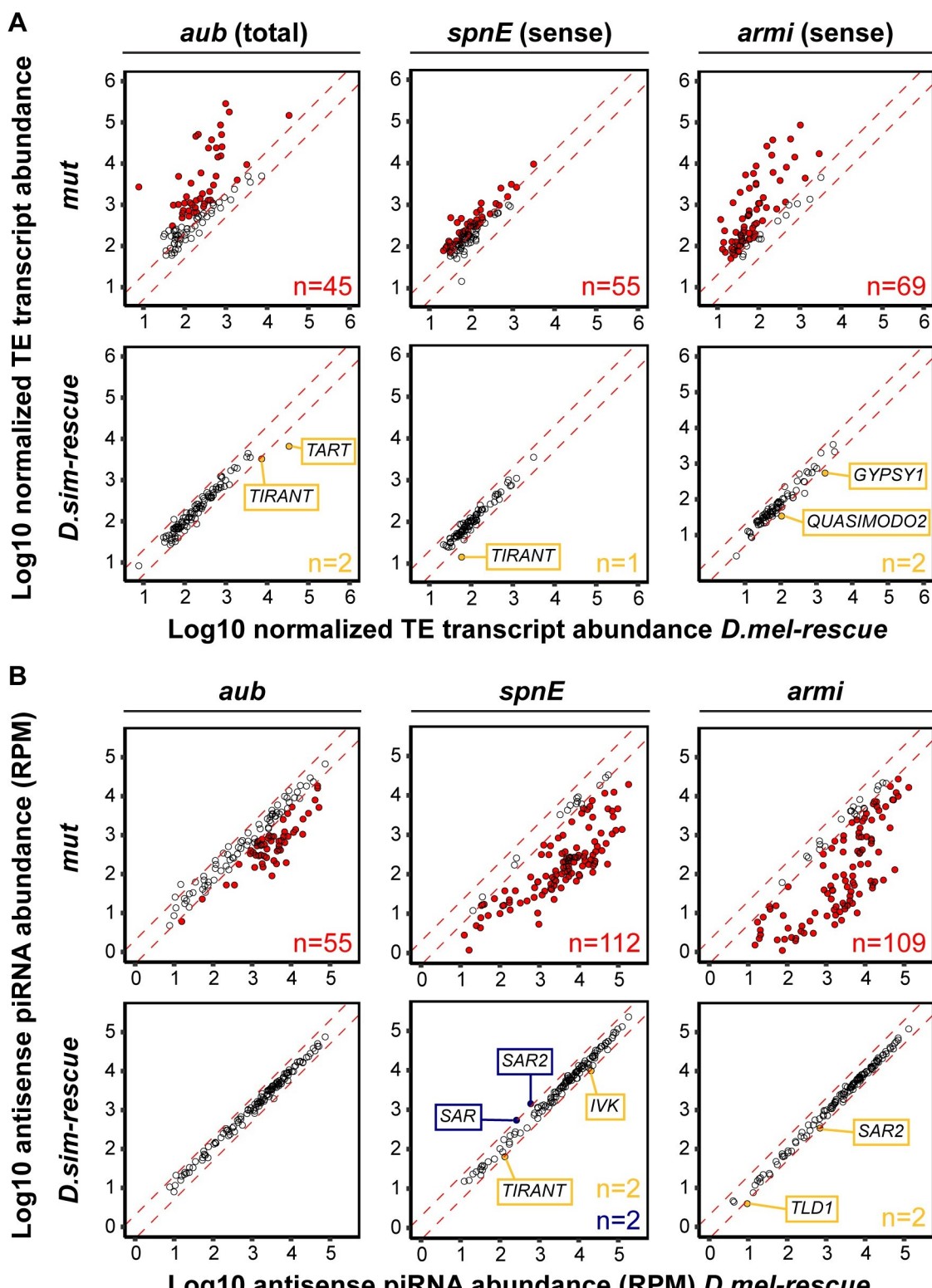

**Fig 2. Minimal differences in TE regulation and piRNA production between alleles.** TE transcript abundance (A) and TE-derived antisense piRNA abundance (B) is compared between *D. melanogaster* rescues and either trans-heterozygous mutants ("*mut*", upper row) or *D. simulans* rescues (lower row) for *aub*, *spn-E* and *armi*. Red dashed lines indicate the two fold-change threshold. TE families whose abundance differs substantially between mutants and *D. melanogaster* rescues are indicated in red ($p < 0.05$ for TE transcripts; >2-fold for piRNA abundance). *P*-values were not considered for small RNA analysis because the small

number of TE families in the analysis (<130 families, S4 Table) is unlikely to provide a sufficiently large sample size for the statistical model implemented in DESeq2 [43]. TE families whose abundance is more than two-fold higher in *D. melanogaster* rescues than in *D. simulans* rescues are in yellow, whereas the reciprocal is in blue. Total mRNA abundance is from unstranded mRNA sequencing of one biological replicate (*aub*), sense RNA abundance is from stranded RNA sequencing of three biological replicates (*spn-E* and *armi*). TE-derived piRNA abundance is based on two biological replicates for *aub* and three biological replicates for *spn-E* and *armi*, and was normalized to the total number of sequenced miRNAs in the same library.

the overall pool of TE-derived piRNAs, with only 10 repeat classes differentially abundant for sense or antisense piRNAs for any of the three pairs of transgenic rescues, four of which are satellite repeats (*HETRP*, *HMR1*, *SAR* and *SAR2*, Fig 2B, S5 and S6 Figs). Interspecific divergence in piRNA production is therefore modest between alleles, with no detectable impact on the regulation of genomic TEs.

### *Drosophila simulans* alleles exhibit reduced piRNA biogenesis

Despite the absence of large-scale interspecific differences in antisense piRNAs that regulate TEs (Fig 2B), we interrogated piRNA pools associated with each of the transgenic rescues for evidence of underlying differences in piRNA biogenesis. We examined molecular signatures of the two major mechanisms of piRNA biogenesis: ping-pong and phasing. Ping-pong biogenesis produces piRNAs through reciprocal cleavage of complementary precursors (Fig 3A) [24,25]. The frequency of ping-pong amplification is therefore estimated by the fraction of piRNAs occuring on opposite strands of the TE consensus whose sequences overlap by 10 bp, a reflection of the cleavage-site preference of the key ping-pong cycle factors Aub and Argonaute-3 (Ago-3, Fig 3A–3D, S5 Table) [24–26]. In contrast, phasing biogenesis occurs through sequential cleavage of a single RNA strand, which is usually antisense [33,34]. Phasing is detected from the fraction of piRNAs whose 3' ends are immediately followed by a uracil residue (+1-U), as well as the frequency of piRNAs from the same strand that are separated by a distance of a single nucleotide (d1), both of which are diagnostic of cleavage by the phasing nuclease Zucchini (Fig 3E–3G, S6 and S7 Tables) [33,34]. In general, ping-pong and phasing are inversely correlated in mutant piRNA pools, because reducing the frequency of one leads to a proportional increase in the other [33,34].

Aub plays a direct role in ping-pong amplification by cleaving piRNA precursors (Fig 3A) [24–26], and *spn-E* is required for the localization of Aub into the perinuclear nuage, where ping-pong occurs [28]. Mutations in either gene therefore cause a complete collapse of ping-pong amplification (Fig 3B, S7A and S7B Fig, S5 Table) [26,32]. Both *D. melanogaster* and *D. simulans aub* and *spn-E* alleles exhibited a dramatic increase in the ping-pong fraction, indicating a conserved role in ping-pong biogenesis (Fig 3B, S7A and S7B Fig). However in the case of *aub*, ping-pong fractions associated with the *D. simulans* transgenic rescue were modestly yet significantly lower than *D. melanogaster*, and there was a corresponding proportional increase in phased piRNA biogenesis (Fig 3B, 3F and 3G, S8 Fig, S6 and S7 Tables), suggesting reduced efficiency of ping-pong. By contrast, *D. simulans spn-E* allele did not reduce ping-pong (Fig 3B, S7B Fig, S5 Table), yet there was a modest but significant increase in the d1 proportion with the *D. simulans spn-E* rescue (Fig 3F, S6 Table), potentially suggesting increased phasing.

Armi promotes the production of phased piRNAs by binding to antisense piRNA intermediates and facilitating their cleavage by the nuclease Zucchini (Fig 3E) [29,33,34]. Both d1 and +1-U are therefore significantly reduced in *armi* mutants (Fig 3F and 3G, S8 Fig, S6 and S7 Tables). While Armi is not involved in ping-pong, phasing produces Aub-bound antisense piRNAs, which are required for ping-pong biogenesis for some TE families [26,44]. Ping-pong fractions are therefore decreased in *armi* mutants for some TE families (Fig 3C and 3D, S7C

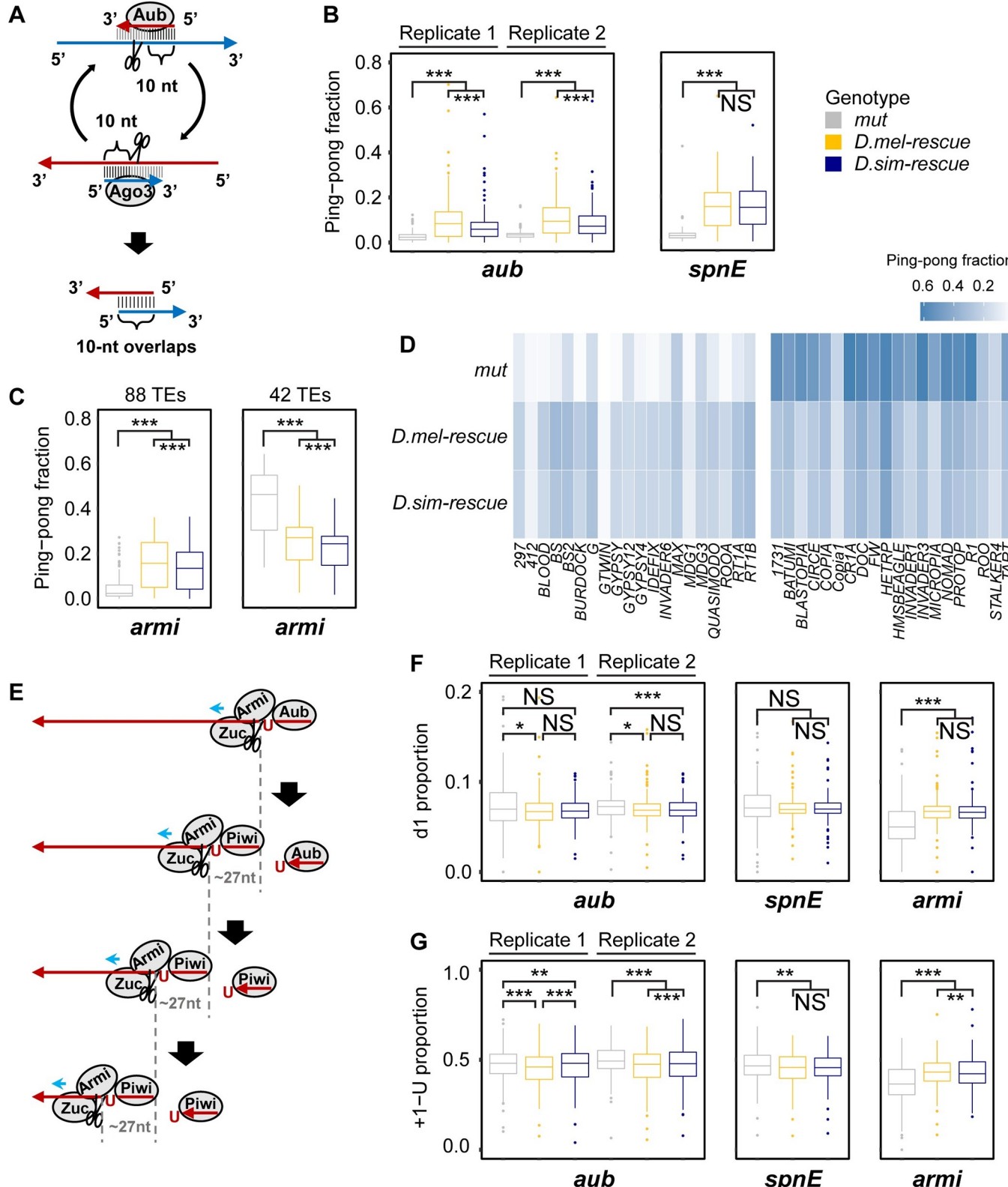

**Fig 3. Drosophila simulans alleles reduce ping-pong biogenesis and phasing biogenesis.** (A) Simplified diagram of ping-pong amplification loop. (B) Ping-pong fractions of TE-derived piRNAs from 142 TE families are compared between trans-heterozygous mutants and transgenic rescues for *aub* and *spn-E*. (C) Ping-pong fractions of TE-derived piRNAs are compared between trans-heterozygous mutants and transgenic rescues for *armi*. Comparison for 88 and 42 TE

families whose ping-pong fractions are decreased (left) or increased (right), respectively, in *armi* mutants as compared to the *D. melanogaster* transgenic rescue. (D) Left: ping-pong fraction heat map for the 20 most piRNA-abundant TE families from panel C left. Right: ping-pong fraction heat map for 20 most piRNA-abundant TE families from panel C right. (E) Diagram of Zucchini-dependent phased piRNA biogenesis. (F) Proportions of 1 nt distance between adjacent piRNAs (d1) mapped to the TE consensus sequences are compared between each genotype of each gene. (G) Proportions of uridine residues immediately after the 3′ ends of piRNAs (+1-U) mapped to the TE consensus sequences are compared between each genotype of each gene. Statistical significance was assessed by the Wilcoxon signed-rank test. In all panels, for *aub*, two biological replicates of each genotype generated at different times are shown separately. For *spn-E* and *armi*, the average of three biological replicates of each genotype generated at the same time are shown. NS denotes $p > 0.05$. $^*$, $^{**}$, and $^{***}$ denote $p \leq 0.05$, $p \leq 0.01$, $p \leq 0.001$, respectively.

Fig, S5 Table). By contrast, for TE families that do not rely on phased piRNA production for ping-pong, ping-pong-derived piRNAs proportionally increase in *armi* mutants, owing to the loss of phased piRNAs (Fig 3C and 3D, S7C Fig, S5 Table). Although exhibiting piRNA production similar to the *D. melanogaster* allele (Fig 2B), the *D. simulans armi* rescue exhibited modestly but significantly reduced +1-U proportion, indicating reduced phasing (Fig 3G, S8B Fig, S7 Table). However, the more dramatic and statistically significant allelic effect is on ping-pong biogenesis, which is reduced for most TE families by the *D. simulans armi* rescue when compared to *D. melanogaster* (Fig 3C and 3D, S7C Fig, S5 Table). Importantly, this reduction occurs regardless of whether *armi* function enhances or represses ping-pong biogenesis, revealing a global inhibitory effect imposed by *D. simulans armi*. Indeed, although the differential abundance of TE and repeat-derived piRNAs between transgenic rescues rarely exceeded two-fold, significantly more TE families were more abundant in the presence of the *D. melanogaster armi* rescue compared to the *D. simulans armi* rescue (118 out of 131 TE families, Sign-test, *P*-value $< 10^{-15}$). Therefore, the modest reductions in ping-pong and phasing biogenesis exhibited by the *D. simulans armi* allele lead to a similarly modest reduction in the abundance of TE and repeat-derived piRNAs.

## Increased off-target effects of *D. simulans* alleles suggest increased genomic auto-immunity

While effective negative regulation of TE transcripts is a critical function of piRNA pathway proteins, it is equally important that they avoid off-target effects that interfere with the function of host genes. [19,21]. Aub, Spn-E and Armi are all RNA binding proteins that must specifically interact with piRNAs, piRNA precursors, and target transcripts, while avoiding interactions with cytoplasmic mRNAs. We therefore considered whether off-target effects differ between *D. melanogaster* and *D. simulans* alleles, predicting that *D. simulans* alleles may produce more off-target effects as they are not adapted to avoid interactions with *D. melanogaster* transcripts.

To test this prediction, we first identified protein-coding genes that are negatively regulated by piRNA pathway proteins by comparing their expression levels between mutants and transgenic rescues (S8 Table). Protein-coding genes whose expression is significantly reduced in transgenic rescues (>1.5 fold) are candidates for off-target effects of piRNA-mediated silencing. We observed that for all three proteins, significantly more genes decreased than increased in expression in transgenic rescues compared to mutants (Fig 4A), suggesting that piRNA pathway proteins tend to reduce the expression of protein-coding genes. Furthermore, the majority of protein-coding genes that are negatively regulated by *D. melanogaster* rescues are also repressed by *D. simulans* rescues, suggesting a shared impact on the expression of many protein-coding genes (Fig 4B and 4C). Indeed, protein-coding genes that are down-regulated by *aub* alleles from either species are enriched among mRNAs bound by Aub (Pearson's Chi-squared test, *P*-value = 0.04) [45].

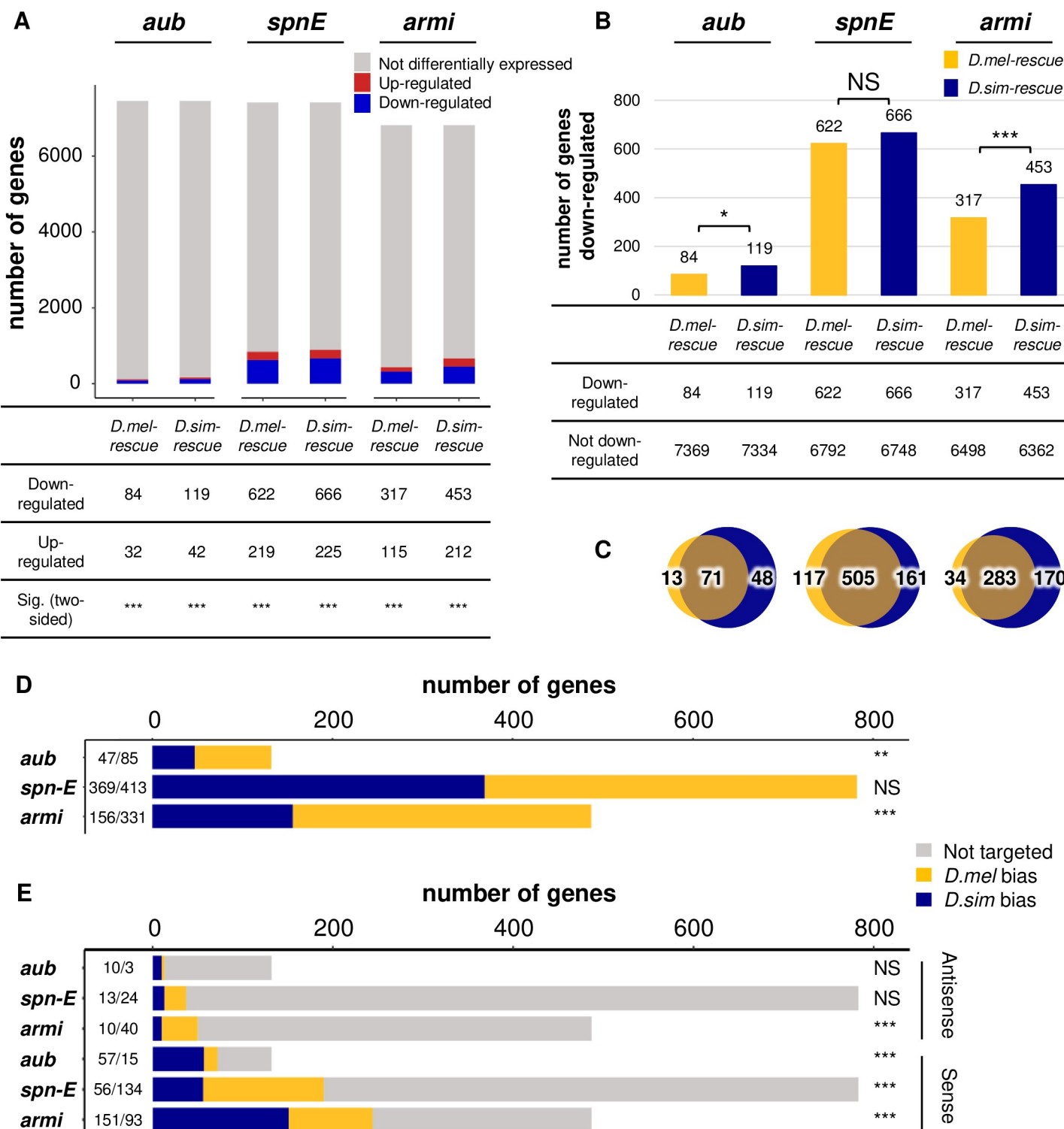

Fig 4. Negative regulation of protein-coding genes suggests increased genomic auto-immunity of *D. simulans* alleles. (A) The number of genes whose expression levels are decreased/increased (>1.5 fold, blue/red) in the presence of each transgene as compared to the corresponding mutant. Statistical significance was assessed by the binomial test evaluating the probability of the observed proportion of down-regulated genes as compared to up-regulated genes under the null hypothesis that the two probabilities are equal. (B) The number of genes whose expression levels are decreased (>1.5 fold) in the presence of each transgene as compared to the corresponding mutant. Contingency tables are shown below. Statistical significance was assessed by the Pearson's Chi-squared Test of Independence. (C) Overlap of genes whose expression levels are decreased (>1.5 fold) in the presence of each transgene as compared to the corresponding mutant for *aub* (left Venn diagram), *spn-E* (middle Venn diagram) and *armi* (right Venn diagram). (D) For genes whose expression levels are down-regulated by alleles from either species, the number of those

whose transcripts are more abundant in *D. melanogaster* rescues than in *D. simulans* rescues is in yellow (log2 fold-change between *D. mel* and *D. sim* > 0), whereas the reciprocal is in blue. Log2 fold-change values of gene expression are based on one biological replicate for *aub* and three biological replicates for *spn-E* and *armi*, and are obtained from a DESeq analysis for *aub* and a DESeq2 analysis for *spn-E* and *armi* (adjusted $p < 0.05$ for Fig 4A–4C). (E) For genes whose expression levels are down-regulated by alleles from either species, the number of those whose antisense/sense piRNAs are more abundant in *D. melanogaster* rescues than in *D. simulans* rescues is in yellow (log2 fold-change between *D. mel* and *D. sim* > 0), whereas the reciprocal is in blue. Genes whose piRNA abundance is too low to estimate the differential expression (< 50 piRNAs on average in at least one genotype) are represented in gray. Genic piRNA abundance is based on two biological replicates for *aub* and three biological replicates for *spn-E* and *armi*, and was normalized to the total number of sequenced miRNAs in the same library. The number of genes corresponding to the blue or yellow part of the bar graph is indicated in Fig 4D and 4E. NS denotes $p > 0.05$. * denotes $p \leq 0.05$. ** denotes $p \leq 0.01$. *** denotes $p \leq 0.001$.

Consistent with the auto-immunity hypothesis, we observed expanded negative regulation of protein coding genes by *D. simulans* alleles of both *aub* and *armi*. While both *D. melanogaster* and *D. simulans* alleles tend to decrease expression of host genes, significantly more genes exhibit reduced expression when the *D. simulans* transgenic rescue is compared to the mutant (Pearson's Chi-squared test, *P*-value = 0.01338 for *aub*; Pearson's Chi-squared test, *P*-value = $4.519 \times 10^{-7}$ for *armi*) (Fig 4B). Furthermore, among those protein-coding genes down-regulated by either transgene, there is a systematic bias towards lower expression in the presence of the *D. simulans* alleles (One-sample Sign-Test, *P*-value = 0.001195 for *aub*; One-sample Sign-Test, *P*-value = $1.332 \times 10^{-15}$ for *armi*). However, the majority of these genes are not significantly differentially expressed between transgenic rescues (127 out of 132 genes for *aub*, 481 out of 487 genes for *armi*) (Fig 4D, S9 Table), indicating that expanded negative regulation by *D. simulans* alleles exhibits only a subtle effect on host gene expression.

Increased off-target effects of *D. simulans* alleles could be explained by increased production of antisense genic piRNAs that target host transcripts, or by piRNA-independent interactions between proteins and mRNAs. Recent analyses of off-target interactions between host mRNAs and Piwi (closely related to Aub) or Armi support the latter scenario, suggesting that while the binding of host mRNAs by piRNA proteins reduces mRNA abundance, it does not result from enhanced antisense genic piRNA production [30,46,47]. We therefore compared the abundance of antisense genic piRNAs that target the silencing of sense transcripts between transgenic rescues (S9 Table). Of 132 and 487 protein-coding genes that are negatively regulated by either allele of *aub* and *armi*, only 13 and 50 are meaningfully targeted by antisense piRNAs, respectively (i.e. >50 antisense piRNAs on average are observed in at least one genotype, Fig 4E). Furthermore, no significant bias towards increased antisense piRNA production in the *D. simulans* allele was observed for either gene; indeed *D. simulans armi* even exhibits decreased, not increased, antisense piRNA abundance when compared to *D. melanogaster* (One-sample Sign-Test, *P*-value = 0.09229 for *aub*; One-sample Sign-Test, *P*-value = $2.386 \times 10^{-5}$ for *armi*) (Fig 4E). Finally, *D. simulans* transgenic rescues do not exhibit expanded production of genic piRNAs for any of the three proteins (S9 Fig). Therefore, enhanced negative regulation of host mRNAs by *D. simulans* Aub and Armi appears to be independent of antisense piRNA production, and may arise through more frequent binding of *D. simulans* proteins to *D. melanogaster* transcripts.

If *D. simulans* Aub and Armi proteins exhibit enhanced binding to *D. melanogaster* mRNAs independently of antisense piRNAs, they could introduce them into the piRNA pool by treating them as substrates for piRNA maturation. Consistent with this model, we observed that sense piRNAs derived from genic transcripts that are negatively regulated by either transgene are significantly more likely to be more abundant in the presence of *D. simulans* alleles of *aub* and *armi* when compared to *D. melanogaster* (One-sample Sign-Test, *P*-value = $6.54 \times 10^{-7}$ for *aub*; One-sample Sign-Test, *P*-value = 0.0002476 for *armi*). However, as with host mRNAs, these increases in sense piRNA abundance are subtle and predominantly not significant for individual genes (71 out of 72 genes in *aub* comparison, and 236 out of 244 genes in *armi* comparison are not significantly different) (Fig 4E, S9 Table). While these modest increases in

sense piRNA abundance are consistent with the use of host mRNAs as substrates for piRNA biogenesis, it is also possible that host mRNAs bound by piRNA proteins may be eliminated by the mRNA degradation machinery [48,49].

## Discussion

Despite pervasive adaptive evolution and gene duplication among piRNA pathway proteins in both insect and vertebrate lineages [12–15,50,51], the underlying forces that drive these evolutionary dynamics remain unclear. By performing interspecific complementation on three adaptively evolving piRNA pathway genes, we revealed diverged functions that may have arisen through positive selection. For all three proteins we observed idiosyncratic differences in TE regulation between *D. melanogaster* and *D. simulans* alleles, which is consistent with genetic conflict between host and parasite. However, we also revealed more extensive off-target effects and reduced efficiency of piRNA maturation associated with *D. simulans* alleles of both *aub* and *armi*, suggesting that selection acts to maximize biogenesis function while minimizing friendly fire on host transcripts. Taken together, our results suggest that positive selection acts at multiple molecular and functional interfaces within the piRNA pathway.

The simplest explanation for the adaptive evolution of piRNA proteins is that selection acts to maximize host control of TE transposition. At face value, TEs that are differentially expressed between transgenes from the two species seem to support this model (Fig 2A). However, all 5 of these TE families were more robustly silenced by the heterospecific *D. simulans* proteins. This suggests that rather than conspecific piRNA proteins being well-adapted to silence their genomic TEs, active genomic TEs may be well-adapted to evade silencing by their host regulators. Indeed the *tirant* element, which is more robustly regulated by *D. simulans* alleles of both *armi* and *spn-E*, is unusually active in *D. melanogaster* but is being actively lost from *D. simulans* [52–54]. We propose that the differential evolutionary dynamics of *tirant* in these two lineages may in part reflect the differences in host-control that we have uncovered.

Genomic auto-immunity was recently proposed as an additional target of positive selection among piRNA proteins [19]. The deliberate non-specificity of piRNA pathway proteins that allows them to target any sequence represented among piRNAs for silencing presents a huge liability for host-gene regulation: how can piRNA proteins avoid deleterious interactions with host transcripts? Furthermore, RNA-immunoprecipitation (RIP) and cross-linking immuno-precipitation (CLIP) of Piwi suggest that piRNA proteins may also negatively regulate host mRNAs by binding them directly in a piRNA-independent manner [46,47]. Similar observations have been made with Armi protein, with the ATP-ase domain being required to disassociate Armi from host mRNAs in the cytoplasm [30]. We observed that *D. simulans armi* and *aub* alleles are characterized by expanded negative regulation of host genes (Fig 4A–4C), which is accompanied by reduced efficiency of TE-derived piRNA production (Fig 3). Importantly, this observation is not consistent with a subtle difference in protein abundance between transgenic rescues, since this would reduce both piRNA biogenesis and off-target effects. Rather our observations suggest that *D. simulans* alleles bind more frequently to host mRNAs (Fig 4), thereby reducing host-gene expression and depleting the pool of protein available to enact piRNA maturation. Nevertheless, we cannot discount an equally intriguing alternative explanation: that *D. simulans* alleles have decreased affinity for interacting protein partners that mediate piRNA biogenesis, which liberates them to bind more frequently to host mRNAs. Future comparisons of molecular interactions involving *D. melanogaster* and *D. simulans* proteins could differentiate between these alternatives.

Our observations considerably expand our understanding of the enigmatic forces that drive adaptive evolution across the piRNA pathway. The three proteins we studied here, which are

cytoplasmic factors involved in piRNA maturation and post-transcriptional silencing, provide an informative contrast to similar studies of three adaptively evolving nuclear transcriptional silencing factors: Rhino, Deadlock and Cutoff [20,22,55]. In comparison to the modest functional differences we observed between *D. melanogaster* and *D. simulans* alleles, nuclear factors are characterized by dramatic interspecific divergence, with *D. simulans* alleles behaving similarly to loss of function or dominant negative mutations [20,22,55]. Furthermore, adaptive evolution among transcriptional silencers has resulted in interspecific incompatibilities between interacting proteins, as opposed to the divergence in protein-RNA interactions that our data suggest. These observations logically reflect differences in the molecular functions of the two classes of proteins, with transcriptional regulation relying on suites of proteins that modify chromatin or regulate RNA-polymerase, while piRNA maturation and post-transcriptional silencing relies more on interactions between proteins and RNA.

Interestingly, a recent meta-analysis of piRNA protein evolution in insects revealed that while positive selection is pervasive throughout the piRNA pathway, signatures of adaptive evolution are significantly stronger among nuclear transcriptional regulators, as compared to the cytoplasmic factors we studied here [14]. Thus, our functional analysis recapitulates an evolutionary signature in sequence data. Why might nuclear transcriptional regulators diverge more rapidly or dramatically than their cytoplasmic counterparts? Enhanced positive selection on nuclear factors may reflect their greater potential to fully suppress the expression of host genes through disrupted chromatin state, as opposed to depleting host transcripts through non-productive binding [19]. We therefore propose that off-target effects may play an under-appreciated role in the evolution of both nuclear and cytoplasmic piRNA proteins, with the strength and consequences of positive selection depending on the mechanisms of—and costs to—host gene regulation.

## Materials and methods

### Fly strains and crosses

All *Drosophila* strains were reared at room temperature on standard cornmeal media.

For the studies of *aubergine* (*aub*), the following *D. melanogaster* strains were used: *w; aub^{N11} bw^1/CyO*, *y w; aub^{HN} bw^1/CyO*, *y w; aub^{HN} bw^1/CyO; ΦP{D. melanogaster aub}*, and *y w; aub^{HN} bw^1/CyO; ΦP{D. simulans aub}*. *w; aub^{N11} bw^1/CyO*, was a gift from Paul MacDonald. *y w; aub^{HN} bw^1/CyO* was obtained by extracting *y w* into *aub^{HN} bw^1/CyO* (Bloomington *Drosophila* Stock Center #8517). *y w; aub^{HN} bw^1/CyO; ΦP{D. melanogaster aub}* and *y w; aub^{HN} bw^1/CyO; ΦP{D. simulans aub}*, originally generated in Kelleher et al [31], were backcrossed for 6 generations in *y w; aub^{HN} bw^1/CyO* to minimize background effects that could lead to differences between transgenic stocks that are unrelated to the transgenes.

For the studies of *spindle-E* (*spn-E*), the following *D. melanogaster* strains were used: *y w; spn-E^1/TM6*, *y w; spn-E^{hls-03987}/TM6*, *y w; spn-E^{hls-03987}/TM6; ΦP{D. melanogaster spn-E}*, and *y w; spn-E^{hls-03987}/TM6; ΦP{D. simulans spn-E}*. *y w; spn-E^1/TM6* and *y w; spn-E^{hls-03987}/TM6* were obtained by crossing *spn-E^1/TM3* and *spn-E^{hls-03987}/TM3* (gifts from Celeste Berg) to *y w; TM3/TM6*. To generate *y w; spn-E^{hls-03987}/TM6; ΦP{D. melanogaster spn-E}* and *y w; spn-E^{hls-03987}/TM6; ΦP{D. simulans spn-E}*, *w^{1118}; ΦP{D. melanogaster spn-E}* and *w^{1118}; ΦP{D. simulans spn-E}* were first crossed to *y w; TM3/TM6*. *+/TM6; ΦP{D. melanogaster spn-E}/+* and *+/TM6; ΦP{D. simulans spn-E}/+* offspring were then crossed to *y w; spn-E^{hls-03987}/TM3*. Finally, *y w; spn-E^{hls-03987}/TM6; ΦP{D. melanogaster spn-E}/+* and *y w; spn-E^{hls-03987}/TM6; ΦP{D. simulans spn-E}/+* offspring were backcrossed into *y w; spn-E^{hls-03987}/TM6* for 6 generations, and subsequently homozygosed for the transgene, to minimize background effects.

For the studies of *armitage* (*armi*), the following *D. melanogaster* strains were used: *y w; armi^1/TM6*, *w; armi^{72.1}/TM6*, *w; armi^{72.1}/TM6; ΦP{D. melanogaster armi}*, and *w; armi^{72.1}/*

*TM6*; Φ*P{D. simulans armi}*. *y w; armi¹/TM6* was obtained by crossing *y w; armi¹/TM3* (Bloomington *Drosophila* Stock Center #8513) to *y w; TM3/TM6*. *w; armi⁷².¹/TM6* was obtained from Bloomington *Drosophila* Stock Center (#8544). To generate *w; armi⁷².¹/TM6; ΦP{D. melanogaster armi}* and *w; armi⁷².¹/TM6; ΦP{D. simulans armi}*, *w¹¹¹⁸; ΦP{D. melanogaster armi}* and *w¹¹¹⁸; ΦP{D. simulans armi}* were first crossed to *y w; TM3/TM6*. *+/TM3; ΦP{D. melanogaster armi}/+* and *+/TM3; ΦP{D. simulans armi}/+* offspring were then crossed to *w; armi⁷².¹/TM6*. Finally, *w; armi⁷².¹/TM3; ΦP{D. melanogaster armi}/+* and *w; armi⁷².¹/TM3; ΦP{D. simulans armi}/+* were backcrossed into *w; armi⁷².¹/TM6* for 6 generations, and subsequently homozygosed for the transgene, to minimize background effects.

Experimental genotypes were obtained from the following crosses. For studies of *aub*, virgin females *w; aubᴺ¹¹ bw¹/CyO* were crossed to (1) *y w; aubᴴᴺ bw¹/CyO*, (2) *y w; aubᴴᴺ bw¹/CyO; ΦP{D. melanogaster aub}* or (3) *y w; aubᴴᴺ bw¹/CyO; ΦP{D. simulans aub}* males. For studies of *spn-E*, virgin females *y w; spn-E¹/TM6* were crossed to (1) *y w; spn-Eʰˡˢ⁻⁰³⁹⁸⁷/TM6*, (2) *y w; spn-Eʰˡˢ⁻⁰³⁹⁸⁷/TM6; ΦP{D. melanogaster spn-E}* or (3) *y w; spn-Eʰˡˢ⁻⁰³⁹⁸⁷/TM6; ΦP{D. simulans spn-E}* males. For studies of *armi*, virgin females *y w; armi¹/TM6* were crossed to (1) *w; armi⁷².¹/TM6*, (2) *w; armi⁷².¹/TM6; ΦP{D. melanogaster armi}* or (3) *w; armi⁷².¹/TM6; ΦP{D. simulans armi}* males. Crosses were maintained at 25˚C on standard cornmeal media.

## Generation of transgenic lines

To introduce *D. melanogaster* and *D. simulans* alleles into *D. melanogaster*, we used ΦC31 integrase-mediated transgenesis system [41], which allows for site-specific integration. To generate transgenes, the gene and flanking regulatory regions of *spn-E* (~9.7Kb, *D. melanogaster* Release 6, 3R:15835349..15845065; *D. simulans* Release 2, 3R:9575537..9585081) [56,57] and *armi* (~6Kb, *D. melanogaster* Release 6, 3L:3460305..3466368; *D. simulans* Release 2, 3L:3357002..3363099) [56,57] were PCR-amplified by using corresponding primers (below) and iProof high-fidelity taq DNA polymerase (Bio-Rad).

*D.mel/D.sim-spn-E* forward primer: ATTGAACGCCGTCTATGCCAAGC

*D.mel/D.sim-spn-E* reverse primer: ACTGTTCGCCATTGCCACAGATTG

*D.mel/D.sim-armi* forward primer: CACCGCTGAAAGATACGCACACG

*D.mel-armi* reverse primer: GCTAGCCTGCGCTTGGGAGTGTTACCATTCG

*D.sim-armi* reverse primer: GCTAGCCTGACCTCGGGAGTGTTACCACTTC

The PCR products were cloned into pCR-Blunt-II-Topo according to manufacturer instructions (Invitrogen). Mutation-free clones were verified by sequencing.

attB-containing constructs used for site-specific integration were generated by subcloning the NotI/BamHI fragment of each *spn-E* TOPO plasmid, and the NotI/NheI fragment of each *armi* TOPO plasmid into NotI/BamHI and NotI/XbaI-linearized pCasper4/attB [58], respectively. *spn-E* and *armi* transgenic constructs were introduced into *D. melanogaster* at the P{CaryP}attP40 site, and site-specific integration of transgenes was confirmed by PCR [59]. The resulting transgenes were made homozygous in *D. melanogaster* *w¹¹¹⁸*. Transgenes are indicated as ΦP{} in genotypes.

## Female fertility

25–35 individual virgin females of each experimental genotype were crossed to two *y w* males on standard cornmeal media at 25˚C. Fresh media and new males were provided every 5 days. The number of progeny from each 5-day period was quantified.

## Small RNA-seq

3-6-day old female ovaries were dissected from each experimental genotype and placed directly in Trizol reagent (Invitrogen), and homogenized. For *aub* genotypes, Illumina small RNA libraries were prepared by Fasteris according to a proprietary protocol that depletes for 2S-RNA. Because the two biological replicates were prepared at different time points (5/13 and 7/13), they were analyzed separately. Small RNA libraries for *spn-E* and *armi* genotypes were prepared as described in [60]. In brief, total RNAs were extracted according to the manufacturer's instructions, and size fractionated on a 12% polyacrylamide/urea gel to select for 18–30 nt small RNAs. Small RNAs were treated with 2S Block oligo (5'-TAC AAC CCT CAA CCA TAT GTA GTC AAG CA/3SpC3/-3'), and were subsequently ligated to 3' and 5' adaptors, reverse transcribed and PCR amplified using NEBNext Multiplex Small RNA Library Prep Set for Illumina. Small RNA libraries were further purified from a 2% agarose gel and sequenced on a Illumina NextSeq 500 at the University of Houston Seq-N-Edit Core.

## RNA-seq

RNA-seq libraries for the studies of *aub* were generated by Weill Cornell Epigenomics Core according to the protocol of [61]. Briefly, total RNA was extracted from the same ovaries as above, and mRNAs were isolated using poly-T Dynabeads (Invitrogen) according to the manufacturer's instructions. Isolated mRNAs were further fragmented using fragmentation buffer (Ambion), ethanol precipitated, and reverse transcribed using Superscript II (Invitrogen) and random hexamer primers. Second-strand synthesis was performed using DNA polymerase I (Promega). cDNA was purified on a MinElute column (Qiagen), repaired with End-IT DNA repair kit (Epicentre), A-tailed with Klenow enzyme (New England Biolabs), and ligated to Illumina adaptors. Ligated cDNA was gel purified with the MinElute gel purification kit (Qiagen), PCR amplified, and gel purified again to make libraries.

RNA-seq libraries for the studies of *spn-E* and *armi* were prepared by using TruSeq Stranded Total RNA Library Preparation Kit for Illumina. 50 bp reads from each library were sequenced on a HiSeq 2000 (Aub and Spn-E) and a HiSeq 2500 (Armi) by the Weill-Cornell Epigenomics Core. RNA-seq and small RNA-seq data sets are deposited under PRJNA494103.

## Bioinformatic analysis of small RNA-seq libraries

3' Illumina adaptors were removed from sequencing reads by Cutadapt [62]. Sequence alignments were made by Bowtie [63]. Contaminating ribosomal RNAs were identified and removed by mapping sequencing reads to annotated ribosomal RNAs from flybase [64]. TE-derived piRNAs and genic piRNAs were identified by aligning sequencing reads ranging from 23–30 nucleotides (nt) to Repbase [65] or protein-coding gene reference sequence from Flybase [64], respectively, allowing for up to 2 mismatches. The number of reads mapped to each TE family or gene were counted using a Linux shell script. Redundant TE families in Repbase were identified by checking sequence identity (those consensus sequences that were >90% identical across >90% of their length were categorized as the same TE family), and reads mapped to multiple redundant TE families were counted only once. Reads mapped to multiple non-redundant TE families were discarded. To identify miRNAs sequencing reads ranging from 18–22 nt were aligned to a miRNA reference sequence from Flybase [64]. TE families or genes with low read count (< 50 on average) in every genotype library were discarded. piRNA counts for each TE family or gene were normalized to the total number of sequenced miRNAs from each library. Normalized values were used for comparisons of the abundance of piRNAs between libraries.

## Bioinformatic analysis of RNA-seq libraries

Removal of ribosomal RNAs, and identification of TE-derived reads was performed as for small RNA libraries (above) except that 3 mismatches were permitted between sequencing reads and TE consensus sequences. Non TE-derived reads were aligned to flybase annotated transcripts in the *D. melanogaster* reference genome (*D. melanogaster* Release 6) [56,64] by TopHat [66], requiring unique mapping. The number of reads from each protein-coding gene were counted using HTseq-count [67]. TE families or genes with low read count (< 50 on average) in every genotype were discarded. Differential expression was estimated concurrently for TEs and protein-coding genes by DESeq for *aub* [68] and DESeq2 for *spn-E* and *armi* [43]. TEs or protein-coding genes were considered differentially expressed if they exhibited an adjusted $P$-value < 0.05 and a fold-change > 2 for TEs and > 1.5 for protein-coding genes.

## Ping-pong fraction

Ping-pong fraction was calculated as described in [69]. In brief, small RNA sequencing reads ranging from 23–30 nt were aligned to TE consensus sequences from Repbase [65], and redundant TE families in Repbase were identified as described above. For each piRNA, the proportion of overlapping antisense binding partners whose 5' end occurs on the 10th nucleotide was determined. This fraction was subsequently summed across all piRNAs from a given TE family, while incorporating the difference in sampling frequency between individual piRNAs. Finally, this sum was divided by the total number of piRNAs aligned to the TE family of interest. For multi-mappers, reads were apportioned by the number of times they can be aligned to the reference.

## Phasing analysis

Small RNA sequencing reads ranging from 23–30 nt were aligned to the Repbase [65], and redundant TE families in Repbase were identified as described above. To calculate the d1 proportion [34], the number of piRNAs whose 5' end was 1–22 nt downstream piRNA was determined for every TE-derived piRNA. The fraction of distances corresponding to 1 nt was then calculated. To calculate the +1-U proportion [34], the nucleotide after the 3' end of each piRNA was determined based on alignment to the Repbase [65]. The frequency of each nucleotide at the +1 position was subsequently summed across all piRNAs from a given TE family, and the proportion of uridine was calculated. For both analyses, multiply-mapping reads were apportioned by the number of times they aligned to the reference.

## Supporting information

**S1 Fig. *Drosophila melanogaster* genotypes and crossing scheme.**
(TIF)

**S2 Fig. Similar expression level of *aub*, *spn-E* and *armi* transgenes, as well as germline and soma specific genes between *D. melanogaster* transgenic rescue and *D. simulans* transgenic rescue.** Fold-change of expression level of *aub*, *spn-E*, *armi*, germline-specific genes and soma-specific genes between *D. melanogaster* transgenic rescue and *D. simulans* transgenic rescue are shown. Fold-change values are based on one biological replicate for *aub* and three biological replicates for *spn-E* and *armi*, and were obtained from a DESeq analysis for *aub* and a DESeq2 analysis for *spn-E* and *armi*. ** denotes $p \leq 0.01$. NS if not labeled.
(TIF)

**S3 Fig. Size distribution and composition of the small RNA pool for each genotype.**
(TIF)

**S4 Fig. *Drosophila melanogaster* transgenes exhibit similar profiles of piRNA biogenesis to heterozygotes.** *Drosophila melanogaster* transgenes and heterozygotes are compared to trans-heterozygous mutants with respect to transcript abundance (A), TE-derived piRNA abundance (B), ping-pong and phasing biogenesis (C). RNA-seq data comparing *aub* heterozygotes and trans-heterozygous mutants is from [70]. Transcript abundance was normalized to the total number of mapped reads of that library. The small RNA-seq data comparing heterozygotes and mutants for *aub*, *spn-E* and *armi* are from [71]. TE-derived piRNA abundance was normalized to the total number of sequenced miRNAs in the same library. Statistical significance was assessed by the Wilcoxon signed-rank test. For *aub*, two biological replicates of each genotype generated at different times are shown separately. For *spn-E* and *armi*, averages of three biological replicates of each genotype generated at the same time are shown. NS denotes $p > 0.05$. *, **, and *** denote $p \leq 0.05$, $p \leq 0.01$, $p \leq 0.001$, respectively.
(TIF)

**S5 Fig. Decoupling between changes in TE transcript abundance and changes in TE-derived piRNA abundance.** Log2 fold-change TE transcript abundance and TE-derived sense/antisense piRNA abundance between two transgenic rescues for the TE families whose TE transcript abundance is substantively different ($> 2$ fold) between two rescues from Fig 2A. Red dashed lines indicate the 2 fold-change threshold.
(TIF)

**S6 Fig. Minimal differences in sense piRNA production between alleles.**
(TIF)

**S7 Fig. Ping-pong fraction heat map for each protein studied.** (A) *aub*, (B) *spn-E* and (C) *armi*. Among (C), 88 and 42 TE families whose ping-pong fractions are decreased (below red line) or increased (above red line), respectively, in *armi* mutant as compared to those in *D. melanogaster* transgenic rescue are shown.
(TIF)

**S8 Fig.** Observed peaks of 1nt distance (A) and +1-U bias (B) among each genotype for each protein studied.
(TIF)

**S9 Fig. Auto-immunity analysis for the genic piRNA profile.** The number of genes whose corresponding total (A) / anti-sense (B) / sense (C) piRNA abundance is increased ($>1.5$ fold) in the presence of each transgene as compared to the mutant. Contingency tables are shown below. Log2 fold-change values were based on two biological replicates for *aub* and three biological replicates for *spn-E* and *armi*, and were obtained from a DESeq2 analysis (adjusted $p < 0.05$). Statistical significance was assessed by the Pearson's Chi-squared test. NS denotes $p > 0.05$. * denotes $p \leq 0.05$.
(TIF)

**S1 Table. Offspring count from the female fertility test.**
(XLS)

**S2 Table. RNA-seq and small RNA-seq library statistics.**
(XLS)

**S3 Table. Normalized abundance and differential expression of TE transcripts.**
(XLS)

**S4 Table. Normalized abundance and differential expression of TE-derived piRNAs.**
(XLS)

**S5 Table. piRNA ping-pong biogenesis signature for TE-derived piRNAs.**
(XLS)

**S6 Table. piRNA phasing biogenesis signature (d1 proportion) for TE-derived piRNAs.**
(XLS)

**S7 Table. piRNA phasing biogenesis signature (+1-U proportion) for TE-derived piRNAs.**
(XLS)

**S8 Table. Protein-coding genes that are differentially regulated by transgenes as compared to the mutant.**
(XLS)

**S9 Table. Log2 fold-change of transcript/piRNA abundance between *D. mel rescue* and *D. sim rescue*, for protein-coding genes whose expression levels are down-regulated by alleles from either *D. melanogaster* or *D. simulans*.**
(XLS)

## Acknowledgments

We are grateful to Shuqing Ji for assistance with cloning of genomic transgenes, and Paul Mac-Donald and Celeste Berg for providing *Drosophila* strains.

## Author Contributions

**Conceptualization:** Daniel A. Barbash, Erin S. Kelleher.

**Data curation:** Luyang Wang, Erin S. Kelleher.

**Formal analysis:** Luyang Wang, Erin S. Kelleher.

**Funding acquisition:** Daniel A. Barbash, Erin S. Kelleher.

**Investigation:** Luyang Wang, Erin S. Kelleher.

**Methodology:** Luyang Wang.

**Resources:** Daniel A. Barbash, Erin S. Kelleher.

**Visualization:** Luyang Wang, Erin S. Kelleher.

**Writing – original draft:** Luyang Wang, Daniel A. Barbash, Erin S. Kelleher.

**Writing – review & editing:** Luyang Wang, Daniel A. Barbash, Erin S. Kelleher.

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
