## [Decision Letter · Decision Letter 0]

9 Sep 2019

Dear Dr Kelleher,

Thanks very much for submitting your manuscript to PloS Genetics.  

I've carefully read the paper and reviews. As you can see below, the reviewers were intrigued by your findings, but would like to see a more thorough analysis of certain points. In general, the reviewers were thorough, and I think careful attention to their comments will improve the paper, particularly where more clarification or analyses are requested to support its main claims.   

In particular, the reviewers suggested several analyses that might provide insight into the difference between the Dsim and Dmel rescue alleles. These include some clarification of analyses in the paper, as well as some sensible suggestions for additional analyses, e.g., an analysis of potential functional divergence due to protein or silent site evolution (which may either provide an alternative explanation or better supports the model in the discussion), and analysis of transposable elements with abundance differences between the species. 

The authors should also provide a fuller justification of their controls for this finding (which is Dmel rescue rather than Dmel wild-type).

The reviewers were particularly curious about insight into the autoimmunity effect of the Dsim rescue allele.  In addition to the reviewers' comments, I had wondered if there were an differences in the characteristics of the genic piRNAs between the mel-armi and sim-armi rescue strains, despite the similar abundances?

Please also pay attention to reviewer 3's request for additional data in the supplement.

Finally, a few minor changes might also make the manuscript easier to understand, particularly the changes to figures suggested by reviewers 3 and 4. In addition to those, I had a few suggestions of my own. 1-The supplementary figure 1 showing which strains are compared would be better as a panel for figure 1 in the main text; otherwise, it's difficult to quickly grasp what comparisons are being made. (Incidentally, this figure's legend is also a little confusing.); 2-In the materials and methods, it would be better to replace the long lists of strains with a table. 

[GPC Note - At least one reviewer noted that "spreadsheets for numerical data underlying graphs, and summary statistics of sRNA and RNA-seq data were not provided". Please note that under PLOS Genetics' data access policy this must be provided - please be sure to attend to this while preparing your revised manuscirpt.]

(Below is the standard boilerplate from the journal; I look forward to seeing a revised version of this manuscript, should you choose to resubmit.)

Thank you very much for submitting your Research Article entitled 'Divergence of piRNA pathway proteins affects piRNA biogenesis and off-target effects, but not TE transcripts, revealing a hidden robustness to piRNA silencing' to PLOS Genetics. Your manuscript was fully evaluated at the editorial level and by independent peer reviewers. The reviewers appreciated the attention to an important problem, but raised some substantial concerns about the current manuscript. Based on the reviews, we will not be able to accept this version of the manuscript, but we would be willing to review again a much-revised version. We cannot, of course, promise publication at that time.

If you decide to revise the manuscript for further consideration at PLOS Genetics, please aim to resubmit within the next 60 days, unless it will take extra time to address the concerns of the reviewers, in which case we would appreciate an expected resubmission date by email to plosgenetics@plos.org.

[LINK]

We are sorry that we cannot be more positive about your manuscript at this stage. Please do not hesitate to contact us if you have any concerns or questions.

Yours sincerely,

Andrea Betancourt

Guest Editor

PLOS Genetics

Gregory P. Copenhaver

Editor-in-Chief

PLOS Genetics

Reviewer's Responses to Questions

**Comments to the Authors:**

Reviewer #1: The fast evolution of piRNA pathway genes has been widely observed, but the underlying evolutionary mechanism has yet to be elucidated. In this study, authors used interspecific (D. melanogaster vs D. simulans) complementation tests of three fast-evolving piRNA pathway genes to investigate this question. While the authors found a significant difference in piRNA biogenesis between mutants with D. melanogaster vs D. simulans transgenes, they found little difference in terms of the TE transcripts. In addition, the authors found that there are more piRNAoff-target effects on host genes in mutants with D. simulans transgene than those with D. melanogaster transgene. Authors concluded that the main driving force for the fast evolution of piRNA pathway genes might come from piRNA production, instead of abundance, and selection for minimizing the piRNA off-target effects on host genes.

This study addresses an important question and could be of wide interests. However, there are some potential technical caveats, and additional analysis/clarifications may be needed to support authors’ conclusions fully. I listed my specific comments below.

1. Potential dosage effect of Aub, Spn-E, and Armi

According to methods and Figure S1, there is only a copy of the transgene in a null mutant background, which is usually used for rescue experiment. However, many piRNA pathway proteins work in concert, and I am wondering whether there could be dosage imbalance that would influence the biogenesis of piRNAs/TE suppression.

2. What is the appropriate baseline?

All the phenotypes assayed (female fertility, piRNA biogenesis, and TE transcripts) are compared to D. mel-rescue. However, it may be necessary also to show that D. mel-rescue has similar phenotypes as those of wildtype/background strains, demonstrating that the D. mel transgene indeed rescue. This issue may be especially important given issues raised in (1). It is possible that single-copy D. mel and D. sim rescues both deviate from the wildtype and this results in little difference observed between D. mel and D. sim rescues.

3. TE families with large divergence in abundance

Although the authors did not find large global differences in TE transcripts between D. mel and D. sim rescues, there are still some families with different transcripts abundance between the two. I am wondering whether these are families that show the largest abundance difference between the two species. It could also be informative to analyze the divergence in piRNA biogenesis with respect to TE copy number difference between the two species.

4. Off-target effects of piRNAs on host genes

What authors observed (fewer protein-coding genes with increased piRNA abundance in D. sim rescue than D. mel rescue) seems to contradict with their findings and interpretations (D. sim rescue leads to more off-target). More explanations/further investigations may be needed. Also, the authors suggest that this phenomenon is similar to what is observed for TEs – no correlation between piRNA biogenesis and expression level. However, an analogy may not be directly drawn here. TE transcripts are also involved in the generation of piRNA biogenesis, while host gene transcripts are not. The underlying mechanisms for the observed phenomena could be very different between TEs and host genes.

5. While I agree with the authors that their finding suggests a threshold effect of piRNA biogenesis (a very important finding!), it is still unclear why piRNA biogenesis may evolve adaptively. Horizontally transferred new TE families/escaped TE families may be one, as proposed by authors, but how the host machinery respond specifically to these families is still unclear. It would be helpful if the authors could further elaborate on this.

Minor comments:

1. Figure 2C – where are these 92/43 TE families in Figure 1? Does this categorization correlate with piRNA abundance?

2. I am a bit confused by Figure 2G, Aub panel. Why the two replicates show different trends (and they are both significant!)?

3. It would help the readers if authors could include a bit more descriptions of piRNA biogenesis through phasing mechanisms.

4. 2x2 tables may be more informative than Figure 5A. (As the Y-axis is not labeled and only described in the figure legend).

5. Figure S2B does not provide enough support that the two transgenes have similar expression level – formal statistical tests are needed. It is also unclear which transgene was chosen as the baseline for calculating fold change (Y-axis).

Reviewer #2: Summary

The authors aim to address the causes and consequences of positive selection on three piRNA pathway genes in Drosophila by creating transgenic rescue lines with D. melanogaster and simulans alleles. The manuscript is well-written and the experiment is an important one, and the results should be of great interest to those studying piRNAs in any species, irrespective of the outcome. Somewhat surprisingly, it appears rescuing D. melanogaster piRNA mutants with D. simulans alleles almost fully restores piRNA pathway function, with seemingly only slight differences in biogenesis and no differences in TE transcript abundance. The authors also find a difference in potential off-target effects – if true, a huge discovery, although more needs to be done to relate this to the other observed phenotypes, as well as confirmatory experiments that Dmel and Dsim proteins are at comparable levels.

Major comments

Overall, more needs to be done to better connect the observed phenotypes and create a clear picture of what the authors propose is happening. This should include both additional analysis (most of which should be able to be mined from available data) and writing, which are outlined separately below.

Analysis/Experiments:

It seems crucial that more should be done to confirm the experiment has worked as expected. For example, are there codon-bias or intron structure differences between melanogaster and simulans that could affect translation? While the authors show similar transcript levels, I think it is imperative that these Dsim and Dmel genes be shown to be translated at similar levels. Even small translation differences in theory could lead to the slightly impaired biogenesis phenotypes observed. Also, figure S2B could use positive controls.

The data presented supporting an autoimmune effect is relatively narrow in scope. Fleshing out this analysis to give a more global picture of what is happening to protein-coding genes during transgene rescue would help readers visualize exactly what is happening. For example, I am also interested in the number of genes that are upregulated by both transgene rescue – if similar numbers to those downregulated, then this may seem more like noisy RNA-Seq data or broader transcriptional dysregulation. If significantly more genes are downregulated upon rescue than upregulated, this would further support the autoimmunity hypothesis. Second, it seems Figure 5 and Figure S6 should be combined into one analysis addressing the question: Does increased off-target effects lead to any meaningful differences in protein-coding gene expression? The authors show that the number of downregulated genes are more than expected (although I cannot find how “Expected” is calculated) and that the number of piRNA producing protein coding genes is more than expected in Dsim rescue, but these figures don’t seem meaningful if there is no overlap. I would also be interested to see how variable piRNA production from CDS is across replicates (for example, does gene X consistently produce more piRNAs upon Dsim rescue, or is a random draw of genes with higher piRNAs from each replicate).

Explanation: A couple of explanations could be provided to better connect the data, and frame it in what is already known or hypothesized. For example, the introduction (starting at P4L20) sets up competing hypotheses for what drives adaptation in piRNA pathway genes. It seems these options should be re-weighed in turn in the discussion. What scenarios is this data consistent with? Yes, various causes of positive selection but also e.g. differences in constraint between species, allowing divergence of neutral variation in Dsim that is functional in Dmel.

Additionally, it is not clear how the authors believe genomic autoimmunity is occurring, and whether it is related to the decreased piRNA biogenesis. It would help readers to either explicitly say that these phenotypes have independent or correlated causes, and to integrate autoimmunity into the Figure 6 summary.

Minor comments

It seems discussing piRNA clusters and their protection would help general interest readers – without this knowledge the conundrum of why piRNA pathway genes are evolving adaptively is less apparent.

Why no ping-pong peak graphs?

It would help general interest readers to include Spn-E in all of the figures where appropriate, and especially in the summary Figure 6.

On P9L19-20, a reference to Figure 2B should be included alongside 2F and G.

I had a very difficult time interpreting Figure 4B, perhaps explain it more thoroughly in the sentence starting on P13L18 “This counterintuitive positive…”, for example something along the lines of “notice how the colors relate in X way with the relationship between LFCs”

It is not clear how expected numbers of autoimmune targets (Figure 5, S6) were calculated, this should be outlined. Also, is the observed significantly different from the expected? I only see a test done for comparing the two observed values, in which case, what are the expected values for?

Discussion P20L9: Typo in melanogaster

Reviewer #3: Please find the review document (Wang_et_al2019_review.docx) attached.

Reviewer #4: Overall, the conclusions of this paper are well supported. However, it would be worth outlining some caveats to these interpretations. In addition, in parts, the presentation of the data could be more clear. In places, the impact of the results is somewhat hidden in the visuals.

1. Could the authors provide an alignment of the protein sequences of the sim and mel alleles? The authors may want to relegate this to the supplemental, but unless these alignments are unwieldy (hopefully they can be presented in a way they are not), I would recommend they put these alignments in the first figure. Without this, the reader is left wondering what kind of divergence there is between the sim and mel protein sequences.

1. One of the main results is that the sim-armi allele leads to a universal reduction of piRNA abundance against TEs. Since this is normalized against total sequence reads, what small RNA class is increased in their place? In the methods, it is stated that TE-piRNA counts were normalized to miRNA. It would be worth reminding the reader that this class was increased as a proportion of total small RNA reads.

2. I feel the scatter plot buries the observation of reduced piRNA amounts. Can a miRNA-normalized size histogram of piRNAs mapping to TEs be provided for each of the alleles? That way, the reader can see the whole distribution presumably downshifted for the sim variants - especially for armi. This would hide some of the TE level variation for sim spnE, but would be a helpful view of total piRNA amounts mapping to TEs. Overall, for TE mapping piRNAs, the reader needs to also have a fold reduction number (50% reduction or whatever) for each of the alleles that they can grasp. Especially since it is against this fold reduction of piRNA that the TE expression will be measured against.

3. Leading into the TE expression section, the authors should remind the reader about the total fold reduction in piRNA abundance for the three sim alleles and ask if there is a proportional increase in TE expression. Again, it would be helpful to get a single metric for fold change in TE expression across the TE bulk (perhaps for just the TE that are misregulated in the mutants stocks). The point I think the authors are trying to make is that a proportional decrease in total piRNA amounts doesn't lead to a proportional increase in TE expression. To make that argument, the figures as presented are not sufficient. They need to present these numbers directly in the text for bulk decrease of TE piRNAs and bulk increase for TE expression.

4. For figure 4B, Dmel-armi vs. Dsim-armi, the scale for log2FC TE transcript abundance should be on the same level as in 4A for spnE and aub. As presented, it gives the impression that the variance for log2FC TE expression is higher for armi. Maybe scale all of them from -1 to 2?

5. Based on Figure 4A, in the text the authors state (page 13, line 13): Strikingly, we observed no correlated changes between piRNA and mRNA pools for aub and spnE when the two transgenic rescues are compared to each other, suggesting the magnitude of changes in piRNA abundance are not sufficient to impact downstream targets.

I feel like the use of the word "Strikingly" is misleading. If one looks at the range of piRNA level change in the mutant vs. transgene that is equivalent in the transgene vs. transgene range, one also doesn't see a relationship in the mutant vs. transgene plots. So, clearly the magnitude of piRNA abundance change is not sufficient in the transgene vs. transgene contrast to reveal this relationship. The way the figure is set up, it might imply to the reader that there is a difference in the relationship between piRNA abundance and expression. But, clearly the results do support the conclusion that TE control is robust to changes in piRNA abundance in this range.

6. Why does 4A examine Log2FC for total TE transcript abundance by 4B look at antisense transcript abundance?

7. Regarding genic piRNAs - perhaps I don't follow Figure 6B, but in a direct contrast of sim vs. mel alleles, are there more genic piRNAs in one vs. the other? 6B seems to be relative to mutant, but what about vs. each other? Perhaps some clarification of this would be helpful.

**Have all data underlying the figures and results presented in the manuscript been provided?**

Reviewer #1: Yes

Reviewer #2: Yes

Reviewer #3: No: Spreadsheets for numerical data underlying graphs, and summary statistics of sRNA and RNA-seq data were not provided.

Reviewer #4: Yes

PLOS authors have the option to publish the peer review history of their article (what does this mean?). If published, this will include your full peer review and any attached files.

Reviewer #1: No

Reviewer #2: No

Reviewer #3: No

Reviewer #4: No

---

## [Decision Letter · Decision Letter 1]

12 Feb 2020

Dear Dr Kelleher,

Thank you very much for submitting your Research Article entitled 'Adaptive evolution among cytoplasmic piRNA proteins leads to decreased genomic auto-immunity' to PLOS Genetics. Your manuscript was fully evaluated at the editorial level and by independent peer reviewers. The reviewers appreciated the attention to an important problem, but raised some substantial concerns about the current manuscript. Based on the reviews, we will not be able to accept this version of the manuscript, but we would be willing to review again a much-revised version. We cannot, of course, promise publication at that time.

If you decide to revise the manuscript for further consideration at PLOS Genetics, please aim to resubmit within the next 60 days, unless it will take extra time to address the concerns of the reviewers, in which case we would appreciate an expected resubmission date by email to plosgenetics@plos.org.

[LINK]

We are sorry that we cannot be more positive about your manuscript at this stage. Please do not hesitate to contact us if you have any concerns or questions.

Yours sincerely,

Andrea Betancourt

Guest Editor

PLOS Genetics

Gregory P. Copenhaver

Editor-in-Chief

PLOS Genetics

Dear Erin:

Many apologies for the delay in getting this back to you. As you noted in your letter, the revisions last time were extensive, and involved substantial reinterpretation of the results. One of the reviewers had substantial remaining concerns, so I read the manuscript and review carefully before making my recommendation.

The reviewers and I found the revised manuscript greatly improved and of broad general interest, particularly the genomic auto-immunity interpretation. I also appreciated the new, more nuanced discussion of the results. There are a few minor presentation issues picked up by the reviewer or myself. There are also a few analyses suggested by reviewer 3: please carefully consider whether these analyses would strengthen the evidence for autoimmunity. While I’ve suggested major revisions, most of these revisions are minor, with the possible exception of the new analysis.

Best wishes,

Andrea Betancourt

• Please clarify whether the data in Figure S3 is total piRNAs, or restricted to those that map to TEs.

• Typo in table associated with figure 4A; ‘reguated’

• Consider replacing Figure 4 panels D and E with analogs of Figure 2B. If not, please improve the figure legend.

• Consider revising Table S9 so that the reader can more easily see the increase in the D. simulans sense piRNAs.

• Consider the analysis of the already available aub-iCLIP data suggested by the reviewer; I agree that, if the data are appropriate, this analysis would strengthen the evidence for promiscuous binding of mRNAs.

• Please take care to italicise ‘D. melanogaster’ and ‘D. simulans’.

• Please note the other minor issues raised by reviewers 2 and 4.

Reviewer's Responses to Questions

**Comments to the Authors:**

Reviewer #2: The newly revised manuscript has addressed my concerns. The new analyses bridging piRNAs and genic transcripts were informative. I found only one technical error:

Figure S3B – 2nd row 2nd column the top of the axis is cut off.

Reviewer #3: The review is uploaded as an attachment

Reviewer #4: This article is greatly improved for clarity and it seems the major concern this reviewer had was resolved by identifying and fixing an artifact in the previous analysis.

My only suggestion is for greater clarity in figures 4D and 4E. The represented probability distributions are estimated from the data, but they should overlay the actual data on these plots. In addition, the following values on the plots should be defined in the figure legend:

1) What is 's'? It looks like the number for LogFC2 >0, but I would recommend deleting that because it is both redundant and confusing.

2) Please clarify that the dotted grey line is the zero point and the dotted blue line is the mean(?)/median(?)..

3) Label the three panels in D and E both with aub, spnE and armi? I assume that is the order, but it is not 100% clear.

**Have all data underlying the figures and results presented in the manuscript been provided?**

Reviewer #2: Yes

Reviewer #3: Yes

Reviewer #4: Yes

PLOS authors have the option to publish the peer review history of their article (what does this mean?). If published, this will include your full peer review and any attached files.

Reviewer #2: No

Reviewer #3: No

Reviewer #4: No

---

## [Editor Report · Decision Letter 2]

20 Apr 2020

Dear Dr Kelleher,

Thank you very much for submitting your Research Article entitled 'Adaptive evolution among cytoplasmic piRNA proteins leads to decreased genomic auto-immunity' to PLOS Genetics.

Thanks for your manuscript, and apologies for the delay in getting back to you. I've [AB] now taken a careful look, and agree that the results will be of general interest to readers of Plos Genetics I do think that the manuscript suffers from a lack of clarity on a few points. None of these problems are major, but I do think a bit more polish and clarity will increase the impact of the work. I'm recommending minor changes in order to give you an opportunity to take care of these problems, which are indeed very minor.

Abstract Line 17: stray 't'

p. 8 "Samples sizes" => Sample sizes

p. 8 Line 8: " Because D. simulans transgenes do not exhibit significantly reduced expression when compared to D.

melanogaster (S2 Fig)": Unless I've misread Figures S2, it shows 'significantly reduced expression' for the D simulans spnE transgene. Please clarify.

p. 9 "This surprising bias towards enhanced negative regulation by D. simulans is not consistent with the random segregation of TE copies during backcrossing."; please clarify what this means.

p. 9, Lines 3 & 15, "5 out of 5 TE families": I think there are only 4 families, as Tirant appears twice.

Fig 2. Why is total TE transcript level shown for aub, rather than only sense as for the other two proteins? Non-stranded libraries? If so, that's fine, but please clarify. Why are sense TE piRNAs never shown?

Fig. S4-- What do the rectangles represent- is it the range between the two? Also, it would be helpful to remind the reader of which two transgenes from figure 2A are being compared/

Please read over the supplementary figure legends, and make sure they are complete.

Please also consider mentioning the (admittedly modest) differences in piRNA biogenesis in the abstract, and consider clarifying that these do not result in global piRNA differences at the beginning of this section, as otherwise the reader is left wondering about this point.

Provide a striking Image with a corresponding caption to accompany your manuscript if one is available (either a new image or an existing one from within your manuscript). If this image is judged to be suitable, it may be featured on our website. Images should ideally be high resolution, eye-catching, single panel square images. For examples, please browse our archive. If your image is from someone other than yourself, please ensure that the artist has read and agreed to the terms and conditions of the Creative Commons Attribution License. Note: we cannot publish copyrighted images.

You can use this link to log into the system when you are ready to submit a revised version, having first consulted our Submission Checklist.

[LINK]

Yours sincerely,

Andrea Betancourt

Guest Editor

PLOS Genetics

Gregory P. Copenhaver

Editor-in-Chief

PLOS Genetics

---

## [Editor Report · Decision Letter 3]

14 May 2020

Dear Dr Kelleher,

We are pleased to inform you that your manuscript entitled "Adaptive evolution among cytoplasmic piRNA proteins leads to decreased genomic auto-immunity" has been editorially accepted for publication in PLOS Genetics. Congratulations!

Yours sincerely,

Andrea Betancourt

Guest Editor

PLOS Genetics

Gregory P. Copenhaver

Editor-in-Chief

PLOS Genetics

Comments from the reviewers (if applicable):

Dear Erin:

Thank you for the revised version of the manuscript, and especially for the increase in clarity on a few points of confusion here and there in the manuscript. I think making the mansucript easier to digest will increase the impact of the work, which is interesting and potentially very important.

Hope all is well.

Best wishes,

Andrea

**Data Deposition**

http://datadryad.org/submit?journalID=pgenetics&manu=PGENETICS-D-19-01131R3

**Press Queries**

---

## [Editor Report · Acceptance letter]

4 Jun 2020

PGENETICS-D-19-01131R3 

Adaptive evolution among cytoplasmic piRNA proteins leads to decreased genomic auto-immunity 

Dear Dr Kelleher, 

We are pleased to inform you that your manuscript entitled "Adaptive evolution among cytoplasmic piRNA proteins leads to decreased genomic auto-immunity" has been formally accepted for publication in PLOS Genetics! Your manuscript is now with our production department and you will be notified of the publication date in due course.

With kind regards,

Jason Norris

PLOS Genetics

On behalf of:
